# Nonresponders of Physical Activity on Prescription (PAP) Can Increase Their Exercise Capacity with Enhanced Physiotherapist Support

**DOI:** 10.3390/ijerph18094795

**Published:** 2021-04-30

**Authors:** Tom Martinsson Ngouali, Mats Börjesson, Åsa Cider, Stefan Lundqvist

**Affiliations:** 1Center for Health and Performance, Department of Food and Nutrition and Sport Science, University of Gothenburg, 405 30 Gothenburg, Sweden; mats.borjesson@gu.se; 2Centrum för Fysisk Aktivitet Göteborg, Region Västra Götaland, 416 68 Gothenburg, Sweden; stefan.lundqvist@vgregion.se; 3Department of Molecular and Clinical Medicine, Institute of Medicine, Sahlgrenska Academy, Sahlgrenska University Hospital/Östra, University of Gothenburg, 416 85 Gothenburg, Sweden; 4Unit of Physiotherapy, Department of Health and Rehabilitation, Institute of Neuroscience and Physiology, Sahlgrenska Academy, University of Gothenburg, 405 30 Gothenburg, Sweden; asa.cider@neuro.gu.se; 5Department of Occupational Therapy and Physiotherapy, Sahlgrenska University Hospital, 413 46 Gothenburg, Sweden

**Keywords:** physical activity, exercise test, physical fitness, cardiorespiratory fitness, primary health care, physical therapists, health behavior

## Abstract

Swedish physical activity on prescription (PAP) is an evidence-based method to promote physical activity. However, few studies have investigated the effect of Swedish PAP on physical fitness, in which better cardiorespiratory fitness is associated with lower risks of all-cause mortality and diagnose-specific mortality. Direct measures of cardiorespiratory fitness, usually expressed as maximal oxygen uptake, are difficult to obtain. Hence, exercise capacity can be assessed from a submaximal cycle ergometer test, taking the linear relationship between heart rate, work rate, and oxygen uptake into account. The aim of this study was to evaluate exercise capacity in the long term, following PAP treatment with enhanced physiotherapist support in a nonresponding patient cohort. In total, 98 patients (48 women) with insufficient physical activity levels, with at least one component of the metabolic syndrome and nonresponding to a previous six-month PAP treatment, were randomized to PAP treatment with enhanced support from a physiotherapist and additional exercise capacity tests during a two-year period. A significant increase in exercise capacity was observed for the whole cohort at two-year follow-up (7.6 W, *p* ≤ 0.001), with a medium effect size (r = 0.34). Females (7.3 W, *p* = 0.025), males (8.0 W, *p* = 0.018) and patients ≥58 years old (7.7 W, *p* = 0.002) improved significantly, whereas a nonsignificant increase was observed for patients <58 years old (7.6 W, *p* = 0.085). Patients with insufficient physical activity levels who did not respond to a previous six-month PAP treatment can improve their exercise capacity following PAP treatment with enhanced support from a physiotherapist during a two-year period. Future studies should include larger cohorts with a control group to ensure valid estimations of exercise capacity and PAP.

## 1. Introduction

Evidence suggests that insufficient physical activity (PA) is associated with premature death and morbidity [1]. It is the fourth leading risk factor of mortality worldwide and remains a global health threat [2]. Several health outcomes can be gained from increased PA, mainly associated with reduced risk of cardiovascular- and metabolic-related diseases and cancers [3,4]. A minimally recommended dose of 150–300 min per week of moderate-intensity aerobic PA, or 75–150 min of vigorous-intensity aerobic PA has been established by the World Health Organization (WHO) [5], and many countries, including Sweden, have embraced the recommendation nationally [6]. 

One major effect of regular PA and/or exercise is the improvement of cardiorespiratory fitness (CRF) [7], which is associated with a lower risk of all-cause mortality and diagnose-specific mortality [8,9,10]. A scientific statement by the American Heart Association (AHA) in 2016 highlights CRF as a clinical vital sign as well as the underuse of corresponding assessments in clinical practice [11]. However, direct measures of CRF, commonly measured as maximal oxygen uptake (VO_2max_), are not easily obtained in clinical practice. Hence, different surrogate measures are used to predict VO_2max_, among which the Ekblom–Bak test [12] and Åstrand test [13] are two common examples from Sweden. Exercise capacity (EC), expressed as work rate measured in watts (W), can be utilized based on a linear relationship between heart rate, VO_2_, and work rate during exercise on a cycle ergometer [14]. 

It is estimated that one in three adults have insufficient levels of PA globally [15]. Meanwhile, a pronounced decline in CRF between 1995 and 2017 was shown in a large cohort study of Swedish workforce adults [16], implicating the need for efficient methods to increase PA and thereby CRF. However, despite the substantial evidence of the importance of PA and CRF, the implementation of efficient long-term PA-promoting interventions in healthcare settings remains difficult [17]. 

Physical activity on prescription (PAP) is one method to promote PA. PAP treatment according to the Swedish model consists of an individually tailored dialogue, individually dosed PA recommendation including a written prescription and organized follow-up, which are in accordance with the Swedish National Board of Health’s recommendations for disease prevention [18]. All licensed health care professionals in Sweden can offer PAP and the method is used as prevention and treatment for a number of conditions [19]. There is growing evidence of Swedish PAP treatment in the promotion of PA, health-related quality of life, and metabolic risk factors [20,21,22,23,24]. A recent systematic review concluded that the Swedish PAP method increases PA levels [24], but it is not yet empirically defined which feature of the method is the most important. However, individualization through a patient-centered approach, as well as follow-ups, are pronounced key factors in the Swedish PAP [25], which differ the method from other internationally established PA-promoting methods showing a less relevant increase in PA [26]. It is not yet thoroughly investigated if PAP can enhance EC. One study found no change in EC after three months of a standardized form of PAP in patients with permanent atrial fibrillation [27]. Another study found that participants who enrolled in PAP treatment following a transient ischemic attack (TIA) improved physical function, assessed by a six-minute walk test [28]. 

In a previously published study [20], physically inactive patients with metabolic risk factors showed significant improvements regarding PA level, metabolic health, and health-related quality of life (HRQoL) after a six-month PAP intervention. However, some patients did not respond adequately to treatment, still showing insufficient levels of PA after six months. The aim of the present study was to evaluate EC with a long-term perspective, following PAP treatment with enhanced support by a physiotherapist (PT) in a nonresponding patient cohort with insufficient levels of PA, despite having previously received a six-month PAP treatment [20]. We hypothesized that physically inactive patients would improve EC after PAP treatment and that the change in EC would not differ depending on age or gender. 

## 2. Methods

### 2.1. Study Design

The present study followed a quasi-experimental design, evaluating baseline and two-year follow-up data. It was an observational substudy of a randomized controlled trial (RCT) [29], which compared a continued ordinary PAP treatment at health care centers (HCC) and PAP treatment with enhanced support from a PT [29]. The RCT was a direct continuation of a previous study that utilized a prospective, longitudinal observational design; its method has been described in detail elsewhere [20]. The present study provided an observational investigation as part of an ongoing study that evaluated Swedish PAP in the short and long term [20,29,30]. 

A total of 195 patients were insufficiently physically active [31] after a six-month PAP treatment [20]. Five of these patients declined further participation, and hence, 190 patients were randomized to either a continued ordinary PAP treatment at HCC (n = 92) or PAP treatment with enhanced support from a PT, which also included testing of EC (n = 98) [29]. In accordance with our aim, only the PT group was further handled in the present study since only the PT group performed EC tests. For detailed information regarding the HCC group and the randomization process, we refer to the RCT [29]. 

### 2.2. Study Population 

A total of 98 patients with insufficient PA levels [31] (<150 min/week of moderate-intensity PA), despite having previously received a six-month PAP treatment [20], and with at least one component of the metabolic syndrome present, were included in the present study. The patient cohort was originally included as a convenience sample from 15 primary HCC in central and western Gothenburg, Sweden, and patients were recruited between 2010 and 2014 [20]. 

### 2.3. Intervention

Recruited patients received PAP treatment with enhanced support and symptom-limited exercise tests from a PT during a two-year period. Appointments and measurements were applied in primary care rehabilitation centers in central and western Gothenburg, Sweden, by PTs experienced in PAP treatment. The individually tailored dialogue had its basis in motivational interviewing (MI) [32]. Following the dialogue, the type and dose of PA were agreed upon according to the patient’s thoughts, possibilities, and the PT’s recommendations. Based on this individualized approach and the result of the exercise test, a prescription of PA was administered. The dialogue was followed by several organized follow-ups, and additional updated prescriptions could be provided. Furthermore, measurements of EC were added to the treatment period, serving both motivational purposes and evaluation. Each patient received a total of 10 appointments with a PT during the two-year period, including measurement occasions at baseline, at four-month follow-up, and at one- and two-year follow-ups. While follow-ups were located within the health care, performed PA occurred outside the health care setting (e.g., brisk walks and exercise at health centers). 

### 2.4. Measurements

Physiotherapists collected data from EC measurements, whereas additional data were collected by personnel at HCC. 

#### 2.4.1. Anthropometrics

A fixed scale (Personmått PEM 136, Hultafors, Sweden) was used to measure body height without shoes in a standing position to the nearest 0.5 cm. Body weight was measured using an electric scale (Carl Lidén, AFW D300, Jönköping, Sweden) to the nearest 0.1 kg. Patients wore light clothing and no shoes. Body mass index (BMI) (kg/m^2^) was calculated accordingly. Waist circumference (WC) was measured using a measuring tape (Kirchner Wilhelm, Aspberg, Germany) to the nearest 0.5 cm. It was placed between the iliac crest and the lower rib on the patient’s skin, with the patient in an upright exhaled position. Systolic and diastolic blood pressure (BP), presented in Table 1, were measured in a seated position after five minutes of rest, using a BP sphygmomanometer (Omron HEM-907, Kyoto, Japan) strapped to the right arm in level with the heart. 

#### 2.4.2. Symptom-Limited Cycle Ergometer Test 

Exercise capacity was measured using a standardized symptom-limited cycle ergometer test. It followed a WHO protocol of utilizing a series of increased loads practiced through stepwise increment, lasting at least four minutes, with a steady state at each level [33,34]. The test was performed using a mechanically braked cycle ergometer (Monark Ergomedic 828E, Vansbro, Sweden). Other materials included a blood pressure cuff (AB Henry Eriksson Diagnostik BS-90, Stockholm, Sweden), a stethoscope, a stopwatch, and a heart rate monitor (Monark, Vansbro, Sweden). The test had a resting phase, work phase, and recovery phase. Systolic and diastolic BP and heart rate (HR) were first measured during rest while the patient was seated on the cycle ergometer. Patients then started the test on a load between 25 and 50 W and were instructed to keep a pedal frequency of 50 revolutions per minute (rpm) during the entirety of the test. During each 4.5-min interval, systolic BP was measured twice (2.5 min and 4.5 min) and HR and rating of perceived exertion (RPE) using the Borg RPE scale [35] were measured twice, respectively (1.5 min and 3.5 min). An increased load of 25–50 W was added onto every new 4.5-min interval. The test was interrupted when patients reached an RPE of 17, if abnormal test results occurred or if a pedal frequency of 50 rpm was not maintained. The time and load for the last interval were then noted. Lastly, systolic and diastolic BP and HR were measured twice, respectively, while patients rested still seated on the cycle ergometer for four minutes. Test results at follow-ups were discussed with patients in relation to previous measurements and reference values [36].

#### 2.4.3. Physical Activity Level

Self-assessed PA was estimated as a means of inclusion to the study. A questionnaire based on the recommendations of the American College of Sports Medicine (ACSM) and AHA [31] assessed if individuals achieved the minimum recommended dose of PA or not. The questionnaire was based on a questionnaire regarding PA that was being validated by the Swedish National Board of Health and Welfare prior to recruitment [37]. Two questions regarding moderate and vigorous PA, respectively, were evaluated, namely, 30 min of moderate-intensity PA per day for each day of the week resulted in one point, and 20 min of vigorous-intensity resulted in 1.7 points. A total score of ≥5 points indicated sufficient levels of PA (≥150 min/week of moderate intensity or ≥75 min/week of vigorous intensity). 

### 2.5. Statistical Analysis

Statistical analysis was performed using SPSS version 25.0 (IBM SPSS Inc., Armonk, NY, USA). Categorical variables were presented as absolute and relative numbers and continuous variables as means and standard deviations (SD) or medians and interquartile ranges (IQR) (Q1–Q3). An alpha level set to <0.05 determined statistical significance. A Wilcoxon signed ranks test was used to evaluate differences between baseline and two-year follow-up data. Mann–Whitney U test was used to analyze differences between subgroups females versus males and dichotomized age groups <58 years old versus ≥58 years old. Age dichotomization was based on the age median for the whole cohort and was used to allow comparisons of two groups of similar size. Subgroup analyses comparing baseline characteristics between the follow-up group versus the dropout group and females versus males were performed using the independent samples t-test or Mann–Whitney U test based on the data level. Effect size (ES) was calculated as r = z/√n [38], where 0.1 = small ES, 0.3 = medium ES, and 0.5 = large ES [39]. Exercise capacity was measured as highest work rate (W_max_) according to Strandell [40]: the heaviest workload in which patients completed 4.5 min of exercise (submaximal power) with an added increment proportional to the completed time (n) of the next added load (X), and was calculated as follows: W_max_ = (submaximal power) + (X · n/4.5)

For example, if a patient completed 4.5 min at 100 W and stopped after two minutes at 125 W, then
W_max_ = 100 + (25 · 2/4.5) = 111.1 W

### 2.6. Ethical Consideration 

Ethical approval was obtained from the Regional Ethical Review Board in Gothenburg, Sweden (Dnr 529-09). Participation was voluntary and patients gave their written consent upon receiving written and verbal study information prior to recruitment. Patients could abort their participation along with collected data at any point. The risk of adverse events following PA in sedentary individuals was acknowledged beforehand [41]. However, Swedish PAP is an established method within the Swedish health care system; PA is associated with major health benefits [3], and patients received guidance from a licensed PT. 

## 3. Results

Overall, 48 patients completed the study, resulting in a dropout rate of 50 (51%) patients. The flow of study participants is outlined in Figure 1, and baseline characteristics are presented in Table 1. No statistical significance was observed when comparing baseline characteristics between the follow-up group and dropout group, and no differences were detected between females and males, except age (mean age 59 years old vs. 54 years old, *p* = 0.040) and WC (mean cm 103.6 vs. 112.6, *p* < 0.001). 

A statistically significant increase in EC was observed at two-year follow-up for the whole patient cohort (*p* ≤ 0.001) with a medium ES (r = 0.34) (Table 2). Subgroups females (*p* = 0.025), males (*p* = 0.018), and age group ≥58 years old (*p* = 0.002) also had a statistically significant increase, whereas age group <58 years old (*p* = 0.085) shows a nonsignificant increase in EC (Table 2). No statistical difference was shown when comparing the mean change of EC between subgroups females versus males (*p* = 0.481) and ≥58 years old versus <58 years old (*p* = 0.788), respectively (Table 3). Prescribed PA and exercise volume are presented in Table 4 and the patients’ individual change in EC is illustrated in Figure 2. 

## 4. Discussion

The main result of this study was the improvement of EC in a nonresponding patient cohort with insufficient levels of PA and metabolic risk factors, following PAP treatment with enhanced support by a PT during two years. The result confirms our hypothesis, as do the similar improvements of EC between subgroups based on age and gender. Although there is a lack of studies specifically investigating EC following PAP treatment, this study follows other studies with positive outcomes regarding Swedish PAP treatment on PA levels [20,21,22,23,24]. 

The findings are considered to have important clinical implications, as they support that a long-term behavior change is possible also in patients with possibly lower levels of motivation, as shown by previous nonresponses to PAP treatment [20]. This long-term approach with enhanced support by a PT appears to promote compliance and affect patients in need of lifestyle change who are difficult to reach by conventional means. The findings are deemed particularly important considering the need for efficient long-term PA-promoting methods [17], the global health threat of lifestyle-related diseases [2], and the declining CRF levels in Sweden [16]. 

Among those patients who improved EC (Figure 2), the response to treatment varied and the mediating factors are not known. In the highly standardized HERITAGE Family Study [42], a cohort of 614 individuals who completed 20 weeks of supervised aerobic exercise consisted of 66% average responders, 13% high responders, and 21% low responders. Heredity was said to have a vital impact. In the present study, however, there is also the factor of compliance to prescribed PA over a two-year period to consider, which was likely limited among patients who decreased EC. In any case, considering the relatively high age of the cohort and the two-year perspective, it is encouraging that the patients who maintained, or only slightly improved EC, at least did not decrease EC since EC usually decreases with increased age among adults [43]. 

Results of the present study showed a statistically significant increase in EC by 7.6 (7.4%) W for the whole cohort after two years with a medium ES (Table 2). This increase is considered clinically relevant based on findings from a recent Swedish study investigating associations between CRF and cardiovascular disease (CVD) morbidity and all-cause mortality in a cohort of 266,109 adults [44]. The study presented risk reductions in CVD morbidity and all-cause mortality by 2.2% and 2.7%, respectively, per every increase in 1 mL·min^−1^·kg^−1^ (working adults, 50–59 years old). These findings are in line with established evidence [8,9,10]. Considering the linear relationship between EC and VO_2_ [14], the relative increase in VO_2_ is approximately the same as the relative increase in EC. Therefore, it is likely that the 7.4% increase in EC is of relevance beyond the statistical significance and ES found in the present study. To exemplify, it means that an individual with a VO_2max_ of 30 mL·min^−1^·kg^−1^ at baseline, with a 7.4% increase posttreatment would have a VO_2max_ of approximately 32.2 mL·min^−1^·kg^−1^, equivalent to estimated risk reductions of CVD morbidity and all-cause mortality by approximately 4.4% and 5.4%, respectively, according to the abovementioned study [44]. These interpretations should be performed with great caution since work rates do not always estimate an accurate corresponding VO_2_, especially at higher work rates, likely due to anaerobic energy metabolism [45]. Unfortunately, calculations on the present data were not possible since weight (kg) variables were not available, whereas relative VO_2_ (mL·min^−1^·kg^−1^) cannot be calculated. 

No difference was found regarding the change of EC between subgroups, neither in absolute nor relative numbers. However, as could be expected, an absolute difference in measured EC was observed, where women and patients ≥58 years old had a lower absolute EC compared to men and patients <58 years old, respectively. As to why the patients <58 years old show nonsignificant results, despite having a similar increase of EC after two years, could be explained by the low cohort size (n = 23) and the presence of outliers within that group, indicated by the relatively large standard deviation (Table 3). 

The dialogue-driven nature of the method and the individualized approach did not allow for a blinding process of patients and therapists. On the contrary, active participation was rather encouraged. This means that a full-on standardization of the treatment was not desirable due to the purpose of individualizing the treatment. The study deviated from a fully individualized experience by standardizing the number of follow-ups and tests for the sake of study procedure. 

Furthermore, the long-term aspect and overall study design did not allow for comprehensive evaluations of performed PA and/or exercise volume. The results presented in Table 4, as well as the statistically significant increase in self-reported PA presented in the main study [29], may provide an insight into the performed volume. It is noted that self-reported PA is associated with certain inaccuracy [46,47], which further exacerbate understanding of compliance and PA and/or exercise volume in this study. These factors are important to highlight because in order to maintain or gain health benefits from PA, including improved CRF, certain volumes are necessary. The ACSM [48] has recommended a minimal volume of ≥30 min/day on ≥5 days/week, resulting in a total of ≥150 min/week of moderate-intensity PA. Alternatively, the ACSM has recommended ≥20 min/day on ≥3 days/week, adding up to ≥75 min/week of vigorous-intensity PA. For deconditioned individuals, lower volumes can also have benefits [48]. The effect of the same volume among individuals may vary due to individual differences in physiological responses to exercise [42,49], which implicates the importance of individualization of prescribed PA, both to promote compliance and regulate prescribed PA. 

A symptom-limited cycle ergometer test was chosen to ensure safe, valid, and objective measurement of EC as well as a basis for prescription. This goes in line with Gappmaier [50], who highlights the importance of an individualized approach among PTs in clinical practice when managing exercise testing. An appropriate exercise evaluation is deemed necessary in order to ensure that the prescribed PA is safe and effective. Proper exercise testing is also of high importance in order to identify patients in need of additional medical attention prior to prescribing PA, e.g., indicated by abnormal test results [50]. There is a wide range of exercise tests to consider. However, the test utilized in the present study was considered appropriate since it allowed for the possibility to detect abnormalities and provided a safe exercise test environment for the patients, who had at least one risk factor of the metabolic syndrome [20]. The test is easier applied and more affordable than direct measurements of CRF, which are key factors in clinical practice. Although the study was not fully reflective of a current everyday approach at rehabilitation centers, the study indicates that a long-term attitude regarding PAP is feasible without requiring large resources. 

### Limitations

Due to various study limitations, certain caution should be taken when interpreting the results. This quasi-experimental study did not have a control group evaluating EC, making it difficult to draw conclusions as to why the group significantly improved EC. A plausible factor could be an increase in PA following PAP treatment [29], which has shown to be an effective method empirically to promote PA among primary care patients [24]. However, considering the lack of control group or associations to other parameters, it is difficult to determine causation to the treatment. Additionally, this study only presented baseline and two-year follow-up data. It would have been beneficial to analyze all EC tests performed to better investigate when the change occurred. Furthermore, the cohort size raises concern for the study’s external validity, especially when considering the overall dropout rate and subgroup analysis. The dropout rate of 51% may influence interpretation and raise concern of selection bias. However, a dropout rate of this magnitude could possibly be expected considering the length of the study and the fact that the participants consisted of a selected cohort of nonresponding patients with metabolic risk factors in a clinical environment. 

## 5. Conclusions

The present study indicates the possibility for a nonresponding patient cohort to improve EC following PAP treatment with enhanced support from a PT during a two-year period. Additionally, our study supports that long-term behavior change is possible in patients with likely lower levels of motivation, as indicated by their previous nonresponse to PAP treatment. This implicates clinical use and further investigations of PAP treatment with enhanced support in order to reach particularly resistant patient groups. The study was unique in the context of PAP research due to its long-term approach. Despite its limitations regarding cohort size, dropout rate, and no control group, future studies should aim for similar long-term investigations. The main goal of PAP is to increase PA levels where improved EC is a major health outcome. This study highlighted EC and its clinical importance and illustrated that it is possible to perform exercise tests within the scope of primary health care. Future studies should apply larger cohorts with a control group to ensure valid evaluations of Swedish PAP treatment. 

## Figures and Tables

**Figure 1 ijerph-18-04795-f001:**
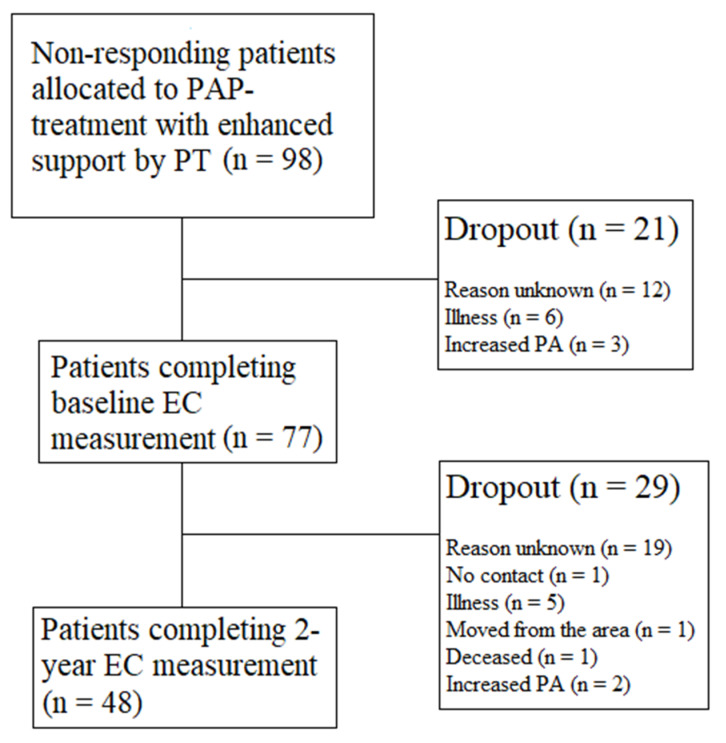
The flow of participants from recruitment to completion of the study.

**Figure 2 ijerph-18-04795-f002:**
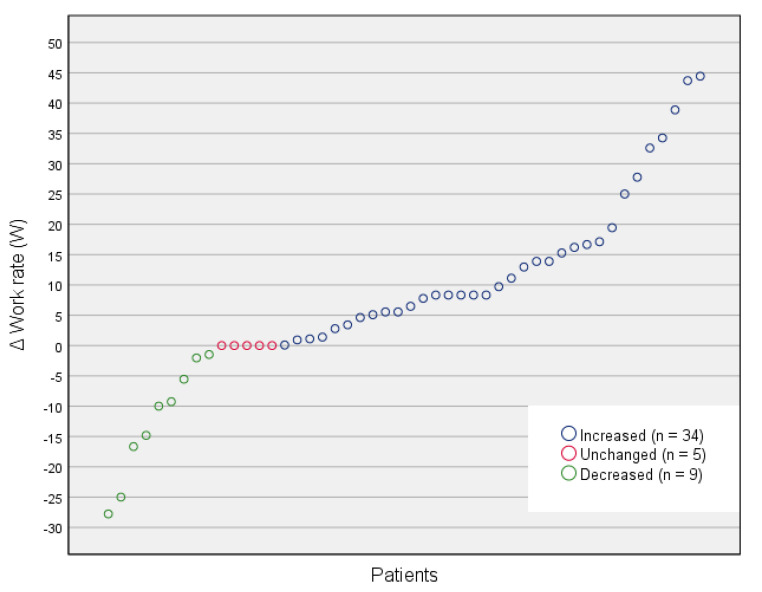
Change in EC (ΔW) for patients completing baseline and two-year follow-up measurements (n = 48).

**Table 1 ijerph-18-04795-t001:** Baseline characteristics of the included patients.

Variable (n)	Value
Age, years (98) ^a^	58.0 (49.0–64.0)
Sex (98) ^b^	
- Female	48 (49)
- Male	50 (51)
PA level, activity points (98) ^c^	2.4 (1.4)
BMI, kg/m^2^ (96) ^c^	32.3 (5.6)
Waist circumference, cm (98) ^c^	108.2 (14.3)
- Female (48)	103.6 (14.1)
- Male (50)	112.6 (13.1)
BP systolic, mmHg (98) ^c^	132.6 (17.1)
BP diastolic, mmHg (98) ^c^	82.2 (10.0)
Smoker (96) ^b^	
- Yes	9 (9.4)
- Former	23 (24.0)
- No	64 (66.6)

Data are presented as ^a^ median (Q1–Q3), ^b^ number (percentage), or ^c^ mean (SD). PA, physical activity; BMI, body mass index; BP, blood pressure.

**Table 2 ijerph-18-04795-t002:** Mean change of measured EC (ΔW) between baseline and two-year follow-up.

Group (n)	Baseline ^a^, Watts	2 Years ^a^, Watts	ΔW ^b^, Watts	*p*-Value ^c^	ES
Total (48)	95.9 (32.3)	103.6 (34.2)	7.6 (7.4)	<0.001	0.34
Females (22)	82.8 (28.8)	90.1 (30.4)	7.3 (8.1)	0.025	-
Males (26)	107.1 (31.4)	115.0 (33.5)	8.0 (6.9)	0.018	-
≥58 years (25)	90.5 (31.9)	98.2 (32.3)	7.7 (7.8)	0.002	-
<58 years (23)	101.9 (32.5)	109.5 (35.9)	7.6 (7.0)	0.085	-

Results are presented as ^a^ mean (SD) or ^b^ mean (%). ^c^ = *p*-value determined by Wilcoxon signed ranks test. Δ, delta; ES, effect size.

**Table 3 ijerph-18-04795-t003:** Mean change of measured EC (ΔW) between baseline and two-year follow-up regarding subgroups.

Subgroup (n)	ΔW ^a^, Watts	*p*-Value ^b^
Females (22)	7.3 (14.0)	0.481
Males (26)	8.0 (16.8)	
≥58 (25)	7.7 (11.0)	0.788
<58 (23)	7.6 (19.4)	

Results are presented as ^a^ mean (SD). ^b^ = *p*-value determined by Mann–Whitney U test. Δ, delta.

**Table 4 ijerph-18-04795-t004:** Prescribed PA- and exercise volume according to the PAP.

Variable (n)	Value
Intensity (44)	
- Vigorous	2 (4.5)
- Moderate	39 (88.5)
- Light	3 (7.0)
Mode (36)	
- AE	21 (58.0)
- RT	0 (0)
- AE + RT	15 (42.0)
Activity (46)	
- Walking	26 (56.5)
- Gym	11 (24.0)
- Biking	4 (9.0)
- Aquatic exercise	3 (6.5)
- Dance/aerobics	1 (2.0)
- Other	1 (2.0)

Results are presented as number (percentage). AE, aerobic exercise; RT, resistance training.

## Data Availability

The data presented in this study are available on request from the corresponding author.

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
