# Peer review of "Nonresponders of Physical Activity on Prescription (PAP) Can Increase Their Exercise Capacity with Enhanced Physiotherapist Support"

_ijerph, 2021, doi:10.3390/ijerph18094795_

Round 1

Reviewer 1 Report

This is a very well written article

I have a few minor comments for your attention:

Please can you state you hypothesis after the aim of the study, and refer to these directly in the discussion. (all participants, gender and age? Given these are how you are presenting statistics?)

“Ninety-eight patients with insufficient levels of PA”  98 out of a possible sample size of how many?  How many declined to participate?

I note this is a sub-study of the RCT: Please clarify the method of randomisation/ The intervention- what happened to the control group?// perhaps useful to clarify this as an observational study arm of the rct? To evaluate repeated measures and assess change in the intervention group only?

Was it a research assistant or the PT who took the study measurements?- please clarify procedure

What is the justification for looking at the age cut off of 58years? (using mean to define the two groups I assume- please define)

Of the approx. half the sample who dropped out, did their participant characteristics differ from those who completed?  Were more of them younger for example (below the 58yrs?), is there something about this group who completed that points towards a targeted approach (clinical implication?).

Reviewer 2 Report

Presented paper is original and presents clearly why PAP can be considered an excellent tool for increase of EC. Authors have themselves written the limitations of the study - especcially that there was no contol group. Maybe the paper could be improved with the short description of the general intenisites of the exercesise which were prescribed  by the physiotherapists. In the discussion maybe it could be more described why authors suggests that there were no responders to the prescribed PE. in this sense maybe the suggetions of physical therapists could be improved for the future. it would be also interesting to see if there is ceartin correlation between different chosen activities and results of responders and non responders to the prescribed activity.

Reviewer 3 Report

This study aims to delineate the role of physiotherapist support in improving compliance and exercise capacity in response to prescribed physical activity. The study population is of interest clinically due to the presence of risk factors for metabolic syndrome and age. Therefore, this study has potential to answer important questions about methods to increase PA in a vulnerable population and thereby improve public health in a meaningful way. The study design does have key limitations, which are acknowledged by the authors, but is nonetheless quite applicable to the research and offers potential insights. Some points mentioned below would strengthen the manuscript.

  • Data such as baseline waist circumference should be broken down by sex. Males and females are known to differ substantially according to sex in clinically important measures such waist circumference and waist to hip ratio. Consistency would also be improved given that the authors reported other results by sex and age subgroups.
  • The authors did not make note of the limitation that compliance with the exercise prescription was only measured by self-report. Self-reported data in factors such as diet and exercise are notorious for inaccuracy. Although the increases in exercise capacity are reasonably expected to be correlated with degree of compliance, we cannot be certain that the participants were accurate in their reporting of the amount of PA accrued. This limitation could have been solved by the use of a fitness watch or pedometer to track the compliance of subjects and would be useful to measure differences in compliance according to whether or not participants receive PT support. This is an improvement to consider in future studies.
  • Change line 286 to "minimal."

Reviewer 4 Report

First of all I want to congratulate the authors for this work. Next, I make a specific report for each section of the article of the aspects that I have reviewed and suggestions for improvement of the article.

Authors should go through an anti-plagiarism program the article and try to lower the percentage that comes out of plagiarism, since it is a bit high.

Title
It is correct and accurately reflects the objective and content of the work.

Abstract
The abstract has a correct scientific structure.

Introduction
The introduction correctly addresses the object of study and contextualizes all the information, but I suggest on page 65 where the authors speak of a Swedish method used to promote the practice of physical activity, it would be advisable to make more information about pre-registration known to the readers of physical exercise in Sweden, which professional does the pre-registration (doctor, Sports Science professional ...), in which cases physical activity is prescribed, etc.

Methodology
The methodology presents sufficient information to understand the research protocol, detailing ethical aspects, instruments used and the research protocol.

Results
The results are clearly presented through different tables and graphs and are related to the objective of the study.

Discussion
The authors make a correct discussion of their results and also address the limitations of their work.

Conclusions
They are correct and consistent with the objective of the article.
In general, the work is scientifically well written and presents important findings, so I believe that it can be published by correcting the couple of suggestions indicated above.
